# Cytokine receptor-Eb1 interaction couples cell polarity and fate during asymmetric cell division

Cuie Chen[1,2,3], Ryan Cummings[1,2], Aghapi Mordovanakis[3], Alan J Hunt[3‡], Michael Mayer[3†], David Sept[3], Yukiko M Yamashita[1,2]*

[1]Life Sciences Institute, Department of Cell and Developmental Biology, University of Michigan, Ann Arbor, United States; [2]Howard Hughes Medical Institute, University of Michigan, Ann Arbor, United States; [3]Department of Biomedical Engineering, University of Michigan, Ann Arbor, United States

**Abstract** Asymmetric stem cell division is a critical mechanism for balancing self-renewal and differentiation. Adult stem cells often orient their mitotic spindle to place one daughter inside the niche and the other outside of it to achieve asymmetric division. It remains unknown whether and how the niche may direct division orientation. Here we discover a novel and evolutionary conserved mechanism that couples cell polarity to cell fate. We show that the cytokine receptor homolog Dome, acting downstream of the niche-derived ligand Upd, directly binds to the microtubule-binding protein Eb1 to regulate spindle orientation in *Drosophila* male germline stem cells (GSCs). Dome's role in spindle orientation is entirely separable from its known function in self-renewal mediated by the JAK-STAT pathway. We propose that integration of two functions (cell polarity and fate) in a single receptor is a key mechanism to ensure an asymmetric outcome following cell division.

DOI: https://doi.org/10.7554/eLife.33685.001

*For correspondence:
yukikomy@umich.edu

Present address: †Adolphe Merkle Institute, University of Fribourg, Fribourg, Switzerland

‡Deceased

## Introduction

Asymmetric cell division is a key mechanism to generate diversity in cell fates. Many stem cells utilize asymmetric cell division to balance stem cell self-renewal and differentiation. Stem cells are often found in a specialized microenvironment, or the niche, that specifies stem cell identity. Asymmetric stem cell division in the context of the niche, thus, involves precisely regulated division orientation with respect to the niche, thereby placing one daughter of the stem cell division inside the niche while the other outside the niche. Despite the clear need of coordinating the niche and stem cell polarity, it remains poorly understood whether and how the stem cell niche may regulate the stem cell polarity. This is largely due to technical difficulties to study stem cell polarity when the stem cell niche function is compromised: in the absence of a functional stem cell niche, the stem cell population is rapidly lost, leaving no stem cells in which the orientation can be examined.

The *Drosophila melanogaster* testis provides an excellent model system for studying asymmetric stem cell division within the niche (*Lehmann, 2012*). *Drosophila* male germline stem cells (GSCs) attach to the hub, a major niche component that secretes the ligand, Unpaired (Upd). Upd binds to Domeless (Dome), a cytokine receptor homolog, leading to activation of the janus kinase-signal transducer and activator of transcription (JAK-STAT) pathway to specify GSC identity (*Kiger et al., 2001*; *Tulina and Matunis, 2001*) (*Figure 1A*). Within the context of this intercellular JAK-STAT self-renewal signaling, GSCs divide asymmetrically by orienting their mitotic spindle perpendicular to the hub (*Yamashita et al., 2003*; *Yamashita et al., 2007*) (*Figure 1A*). Spindle orientation is precisely prepared during interphase by stereotypical orientation of the mother and daughter centrosomes

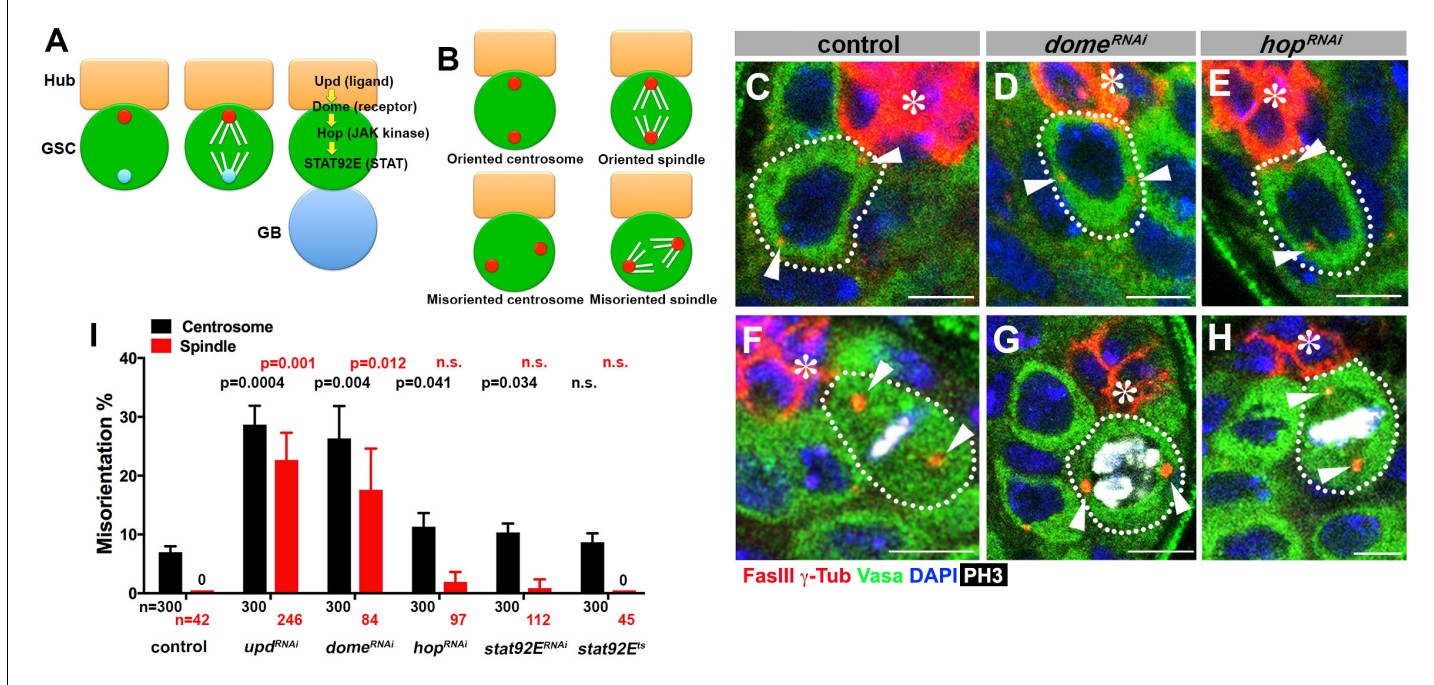

**Figure 1.** *upd* and *dome* regulate centrosome/spindle orientation independent of the self-renewal pathway. (**A**) Asymmetric GSC divisions. Stereotypical positioning of mother (red circle) and daughter (blue circle) centrosomes leads to spindle orientation that places the gonialblast (GB) away from the hub. (**B**) The definition of oriented/misoriented centrosomes/spindles. (**C–E**) Examples of centrosome orientation in control (**C**), *nos-gal4ΔVP16, gal80^ts* > UAS-*dome^RNAi* (4 d after RNAi induction) (**D**), and *nos-gal4ΔVP16, gal80^ts* > UAS-*hop^RNAi* (4 d after RNAi induction) (**E**) GSCs (indicated by a white dotted line). Asterisk indicates the hub. Arrowheads indicate centrosomes. Green: Vasa (germ cells). Red: Fas III (hub cells) and γ-Tubulin (centrosome). Blue: DAPI. Bar: 5 µm. (**F–H**) Examples of spindles in control (**F**), *nos-gal4ΔVP16, gal80^ts* > UAS-*dome^RNAi* (4 d after RNAi induction) (**G**), and *nos-gal4ΔVP16, gal80^ts* > UAS-*hop^RNAi* (4 d after RNAi induction) (**H**) GSCs (indicated by a white dotted line). Arrowheads indicate spindle poles. Green: Vasa. Red: Fas III and γ-Tubulin. White: Thr 3-phosphorylated histone H3 (PH3) (mitotic chromosomes). Blue: DAPI. Bar: 5 µm. (**I**) Summary of GSC centrosome/spindle misorientation in the indicated genotypes. P value comparing control and the indicated genotypes was calculated using two-tailed Student's t-test. Error bars indicate the standard deviation. N = GSC number scored for centrosome orientation or N = mitotic GSC number scored for spindle orientation.

DOI: https://doi.org/10.7554/eLife.33685.002

The following figure supplement is available for figure 1:

**Figure supplement 1.** Validation of RNAi for the JAK-STAT pathway components.

DOI: https://doi.org/10.7554/eLife.33685.003

(*Figure 1A*). This spindle orientation allows one daughter of the GSC division to remain attached to the hub to self-renew, while the other is displaced away from the hub to initiate differentiation.

Here, we show that the receptor Dome plays dual roles in activating the JAK-STAT pathway for GSC self-renewal and orienting the GSC spindle to allow asymmetric stem cell division. We show that these two functions are entirely separable and the spindle orientation is mediated by Dome's direct interaction with the microtubule regulator Eb1. Finally, we show that cytokine receptor-Eb1 interaction is evolutionarily conserved, with a mammalian cytokine receptor, Gp130, regulating the centrosome orientation toward a model immunological synapse. Taken together, we propose a novel mechanism by which a single receptor couples cell polarity with cell fate to ensure obligatory asymmetric division.

## Results

### Niche ligand Upd and receptor Dome regulate spindle orientation during asymmetric divisions of the *Drosophila* male GSCs

To begin to address the potential role of the niche signaling in the oriented stem cell divisions in *Drosophila* GSCs, we first examined whether the JAK-STAT pathway components [*upd* (ligand), *dome* (receptor), *hop* (JAK kinase), *stat92E* (STAT)] might regulate GSC centrosome/spindle orientation in addition to their known role in supporting GSC self-renewal.

Because JAK-STAT components are essential for early development and GSC maintenance, we took advantage of temporarily controlled RNAi-mediated knockdown: we combined *upd-gal4* or *nos-gal4ΔVP16* with *tub-gal80$^{ts}$* to drive the expression of *UAS-RNAi* constructs for the components of the JAK-STAT pathway (*upd, dome, hop* and *stat92E*) in a temporary controlled manner. *UAS-RNAi* construct is not expressed at 18°C, but its expression is induced upon shifting to 29°C (see Materials and methods for details). Expression of RNAi constructs of any JAK-STAT pathway components led to a clear reduction in the STAT level in GSCs by 4 days, and complete GSC loss by 10 days after temperature shift to 29°C (*Figure 1—figure supplement 1*). These results validate the efficiency of RNAi-mediated knockdown of JAK-STAT components. We initially focused on day four after induction of RNAi, when downregulation of STAT is clear but GSC loss is incomplete (~5 GSCs/testis after 4 days of RNAi induction, compared to ~9 GSCs/testis in control testis), such that centrosome/spindle orientation can be assessed in the remaining GSCs (see below for the results of more complete knockdown). Correct centrosome orientation is defined as at least one centrosome being near the hub-GSC junction in interphase, whereas correct spindle orientation is defined as one spindle pole being juxtaposed to the hub-GSC junction in mitosis (*Figure 1B*). Conversely, misoriented centrosomes/spindles are defined as neither of centrosomes/spindle poles being near the hub-GSC junction (*Figure 1B*). In wild type/control testes, most interphase GSCs (>90%) showed correct centrosome orientation, and mitotic spindles were almost always oriented perpendicular to the hub, as reported previously (*Figure 1C,F,I*) (*Yamashita et al., 2003*). Upon knockdown of *upd* or *dome* (*upd-gal4, tub-gal80$^{ts}$ > UAS-upd$^{RNAi}$* or *nos-gal4ΔVP16, tub-gal80$^{ts}$ > UAS-dome$^{RNAi}$* after 4 days of RNAi induction), GSCs exhibited a dramatic increase in misoriented centrosomes and spindles (*Figure 1D,G,I*), indicating that Upd and Dome are critical for centrosome/spindle orientation. Surprisingly, downregulation of JAK kinase (*hop$^{RNAi}$*) or STAT transcription factor (*stat92E$^{RNAi}$*, or temperature sensitive *stat92E$^{ts}$*) barely affected either centrosome or spindle orientation (*Figure 1E,H–I*). The lack of effect on centrosome/spindle orientation in *hop$^{RNAi}$* or *stat92E$^{RNAi}$* is unlikely due to inefficient RNAi knockdown, because *hop$^{RNAi}$* and *stat92E$^{RNAi}$* led to reduction in STAT level in GSCs, followed by complete GSC loss, with the same kinetics as *upd$^{RNAi}$* and *dome$^{RNAi}$* (*Figure 1— figure supplement 1*) (see below for further support of this notion).

These results indicate that *upd* and *dome* might regulate GSC centrosome/spindle orientation independent of the transcriptional network regulated by the JAK kinase and STAT transcription factor.

### Function of Upd and Dome in GSC centrosome/spindle orientation is separable from their role in GSC self-renewal

Because the niche signaling is essential for the GSC self-renewal, above results that *upd* and *dome* are required for the GSC centrosome/spindle orientation may merely reflect that GSC identity is required for centrosome/spindle orientation. However, we disfavor this possibility, because *hop* and *stat92E*, which are equally essential for GSC identity, did not appear to be required for GSC centrosome/spindle orientation (*Figure 1I*).

We were able to separate the function of *dome* in GSC self-renewal and centrosome/spindle orientation by introducing gain-of-function mutation of JAK kinase (*hop*) in the *dome$^{RNAi}$* background (*hop$^{tum-1}$, nos-gal4 > UAS-dome$^{RNAi}$*). Because *dome*'s function in GSC self-renewal is to activate JAK kinase, introduction of a gain-of-function mutation of JAK kinase eliminated the necessity of the receptor Dome in GSC self-renewal. Although *dome$^{RNAi}$* (*nos-gal4 > UAS-dome$^{RNAi}$*, without temporal control of RNAi) resulted in near complete loss of GSCs by the time of eclosion, introduction of *hop$^{tum-1}$* to this genotype background (*hop$^{tum-1}$, nos-gal4 > UAS-dome$^{RNAi}$*) significantly rescued the GSC loss phenotype ((9.1 ± 0.3 GSCs/testis in wild type control, 0.3 ± 0.1 GSCs/testis in *dome$^{RNAi}$*,

$4.9 \pm 0.2$ GSCs/testis in $hop^{tum-1}$ $dome^{RNAi}$, **Figure 2A–C**). This genotype allowed us to examine the role of Dome in GSC centrosome/spindle orientation without being obscured by the loss of GSC identity. We found that, despite that GSCs were well maintained, GSC centrosome/spindle were highly misoriented (**Figure 2D–F**). These results demonstrate that the centrosome/spindle orientation defect due to inactivation of *upd* or *dome* did not arise from the loss of GSC identity. Taken together, we conclude that *upd* and *dome* regulate GSC centrosome/spindle orientation independent of the self-renewal pathway.

It should be noted that GSC centrosome misorientation does not normally lead to spindle misorientation due to a GSC-specific checkpoint mechanism, the centrosome orientation checkpoint (COC) (*Cheng et al., 2008*; *Venkei and Yamashita, 2015*). The COC monitors the correct centrosome orientation and prevents mitotic entry upon sensing centrosome misoreintation. The fact that

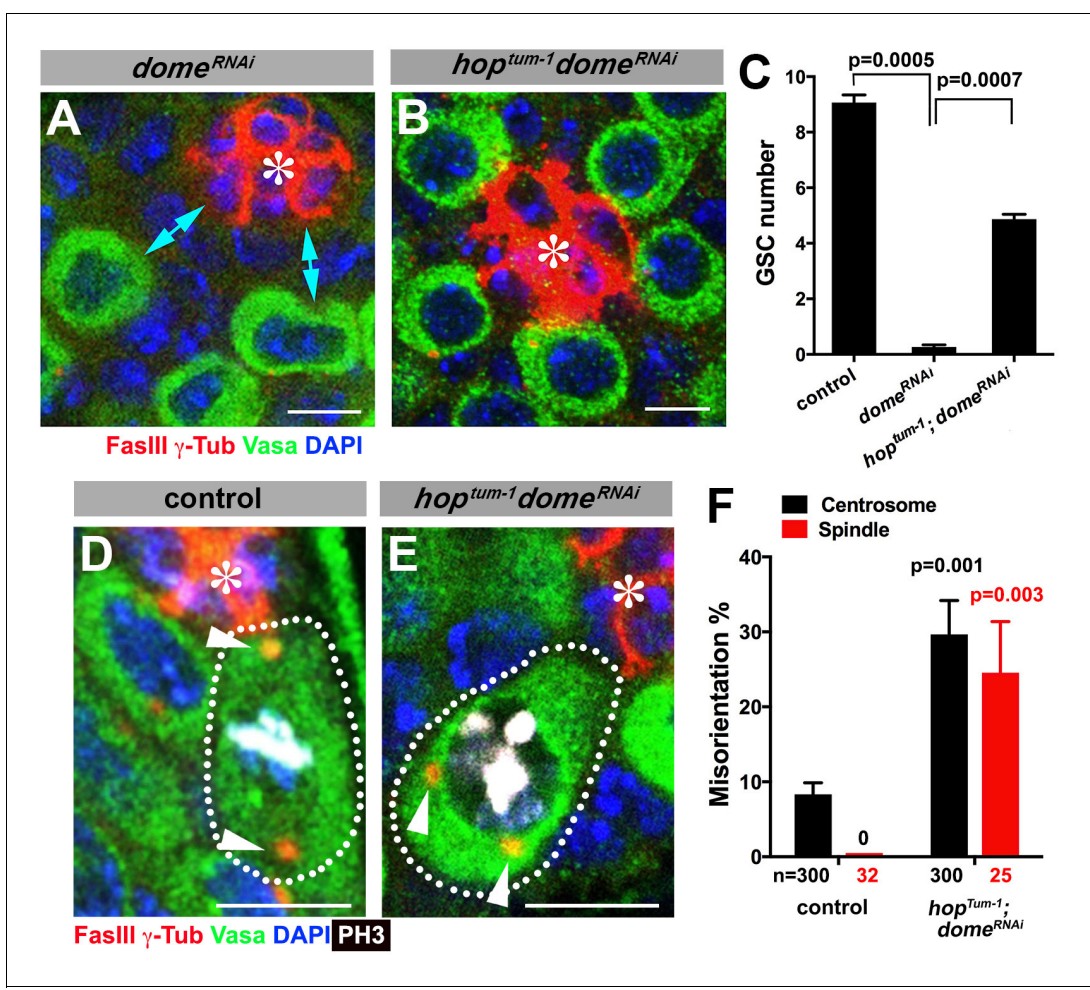

**Figure 2.** Gain-of-function mutant of the JAK kinase *hop* significantly rescues GSC loss but not centrosome/spindle orientation caused by $dome^{RNAi}$. (**A–B**) Examples of apical tip in $nos$-$gal4 > UAS$-$dome^{RNAi}$ (**A**) and $hop^{tum-1}$, $nos$-$gal4 > UAS$- $dome^{RNAi}$ (**B**) testes. Double-headed arrows indicate the gap between the hub and nearest germ cells, showing GSC loss. Green: Vasa. Red: FasIII. Blue: DAPI. Bar: 5 µm. (**C**) GSC numbers in the indicated genotypes. P value was calculated using two-tailed Student's t-test. Error bars indicate the standard deviation. (**D–E**) Examples of spindles in control (**D**) and $hop^{tum-1}$, $nos$-$gal4 > UAS$-$dome^{RNAi}$ (**E**) GSCs (indicated by a white dotted line). Asterisk indicates the hub. Arrowheads indicate spindle poles. Green: Vasa. Red: Fas III and γ-Tubulin. White: PH3. Blue: DAPI. Bar: 5 µm. (**F**) Summary of GSC centrosome/spindle misorientation in the indicated genotypes. P value comparing to control was calculated using two-tailed Student's t-test. Error bars indicate the standard deviation. N = GSC number scored for centrosome orientation or mitotic GSC number scored for spindle orientation.

DOI: https://doi.org/10.7554/eLife.33685.004

*upd*$^{RNAi}$ and *dome*$^{RNAi}$ lead to high frequency of spindle misorientation indicates that Upd and Dome are also required for COC, although the underlying mechanism remains elusive (see Discussion for more detail).

## Dome localizes to the hub-GSC interface during interphase and translocates to the spindle during mitosis

To gain insights into the underlying mechanism by which Upd and Dome may regulate centrosome/spindle orientation, we examined the subcellular localization of Dome during GSC cell cycle. By using a specific anti-Dome antibody (*Figure 3—figure supplement 1*), we found that Dome localizes near the hub-GSC interface during interphase (*Figure 3A*), where centrosome is known to be anchored by microtubules (MTs) (*Yamashita et al., 2007*), implying that Dome might function in anchoring centrosomes to the hub-GSC interface. Dome localization at the hub-GSC interface was dependent on Upd (*Figure 3D,G*), indicating that Upd might guide the localization of Dome. Although Upd is a secreted ligand, it does not diffuse far after secretion (*Harrison et al., 1998*), thus locally concentrated Upd might instruct Dome localization in GSCs. Consistent with the idea that Upd directly guides Dome localization, expression of Upd from a single somatic cyst stem cell clone was sufficient to ectopically localize Dome in the neighboring GSCs (*Figure 3I*, n = 18 clones were examined).

In mitosis, Dome translocated to the spindle (*Figure 3B–C*), which was also dependent on Upd (*Figure 3E–F,H*), again suggesting possible involvement of Dome in MT-dependent processes. Interestingly, although Dome localized to the mitotic spindle even in non-GSCs (i.e. gonialblasts (GBs) and spermatogonia (SGs)), it was not dependent on Upd in these cells (*Figure 3H*), indicating that Upd regulates Dome's localization specifically in GSCs. Importantly, Dome localization was not affected in *hop*$^{RNAi}$ or *stat92E*$^{RNAi}$ (*Figure 3—figure supplement 2*), suggesting that only Upd, but not the JAK-STAT transcription network, is critical for Dome localization. Taken together, these results support the notion that Dome may directly regulate GSC centrosome/spindle orientation, possibly via its interaction to the cell cortex and/or MTs.

These observations raised two critical questions. First, how does Dome, a transmembrane receptor, localize to spindle poles and spindles during mitosis? Second, how does Dome regulate GSC spindle orientation? These questions are addressed in the following sections.

## Dome is endocytosed via early/recycling endosomes in regulating GSC spindle orientation

First, we addressed how Dome might translocate from the hub-GSC interface to the spindle. We hypothesized that Dome might be endocytosed to localize to the spindle. Indeed, previous reports have demonstrated that Dome can be trafficked through the endocytic pathway in several *Drosophila* cell lines and tissues (*Devergne et al., 2007*; *Vidal et al., 2010*). To determine the potential role of endocytic pathways in Dome localization in GSCs, we first examined potential colocalization of Dome with various endocytic Rab GTPases (*nos-gal4 > UAS-YFP-rab*). Obvious colocalization was observed between Dome and early endosomal marker Rab5 specifically in prophase (*Figure 4A–C*). Consistently, endogenous Rab5 was co-immunoprecipitated with Dome using *Drosophila* testes lysates enriched with GSCs, confirming their physical interaction (*Figure 4D*, *nos-gal4 > UAS-dome-GFP*, *UAS-upd*). In addition, a recycling endosome marker, Rab4, colocalized with Dome during prophase and physically interacted with Dome (*Figure 4—figure supplement 1A,F*). Rab8 showed weak colocalization during prophase, but clear physical interaction between Rab8 and Dome was detected (*Figure 4—figure supplement 1C,H*). A recycling endosome marker Rab11 showed colocalization with Dome at prophase, but physical interaction was barely detectable (*Figure 4—figure supplement 1D,I*). A late endosomal marker Rab7 and a recycling endosmal marker Rab35 showed no colocalization or physical interaction (*Figure 4—figure supplement 1B,E,G*).

These interactions between Dome and some endocytic components appear to be relevant, because expression of dominant-negative forms of Rab4, 5, 8, and 11 (*nos-gal4 > UAS-YFP-rab*$^{DN}$) perturbed localization of Dome (*Figure 4—figure supplement 2*) and significantly increased centrosome/spindle misorientation (*Figure 4E,F,G,I,J,L*). Interestingly, expression of Rab4$^{DN}$ or Rab5$^{DN}$ led to perturbation of Dome localization even in interphase, implying that endocytosis is required for Dome localization even in interphase. In contrast, expression of Rab7$^{DN}$ or Rab35$^{DN}$ showed little

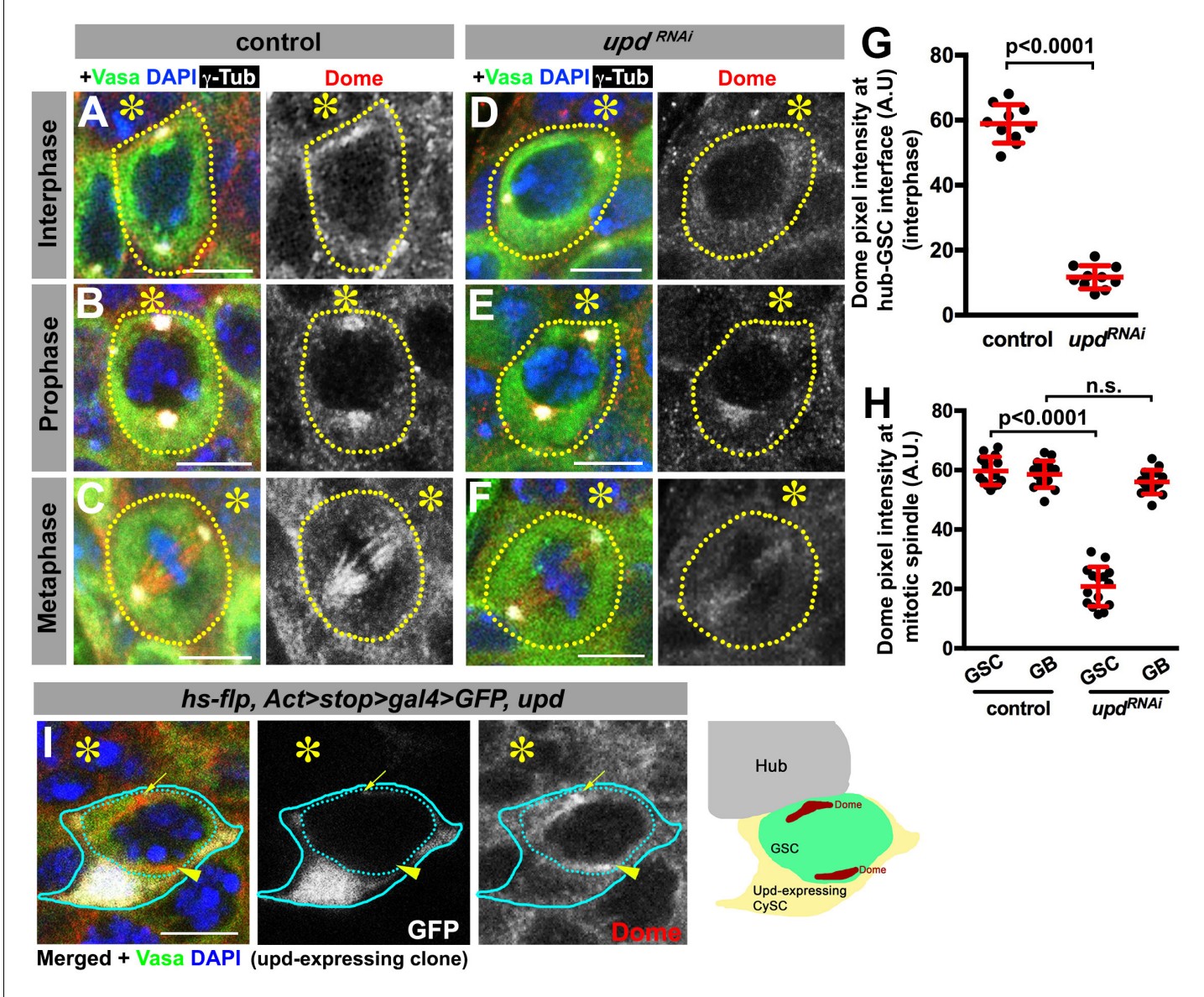

**Figure 3.** Dome localizes to the hub-GSC interface and mitotic spindle in a Upd-dependent manner. (A–F) Dome localization in control interphase (A), prophase (B), metaphase (C), *upd-gal4, gal80^{ts} > UAS-upd^{RNAi}* (4 d after RNAi induction) interphase (D), prophase (E), and metaphase (F) GSCs. Green: Vasa. Red: Dome. White: γ-Tubulin. Blue: DAPI. Asterisk indicates the hub. (G–H) Pixel intensity analyses of Dome at interphase (G) and mitotic (H) GSCs in control or *upd-gal4, gal80^{ts} > UAS-upd^{RNAi}* (4 d after RNAi induction) testes. P value was calculated using two-tailed Student's t-test. Error bars indicate the standard deviation. (I) A GSC (inside the dotted blue line) adjacent to a Upd-expressing cyst stem cell clone (blue line) with ectopic Dome localization (arrowhead), in addition to normal Dome localization near the hub (arrow). Fly genotype: *hs-flp, Act-FRT-stop-FRT-gal4 > UAS-GFP, UAS-upd*.

DOI: https://doi.org/10.7554/eLife.33685.005

The following figure supplements are available for figure 3:

**Figure supplement 1.** Validation of Dome antibody.
DOI: https://doi.org/10.7554/eLife.33685.006

**Figure supplement 2.** JAK or STAT is not required for correct Dome localization.
DOI: https://doi.org/10.7554/eLife.33685.007

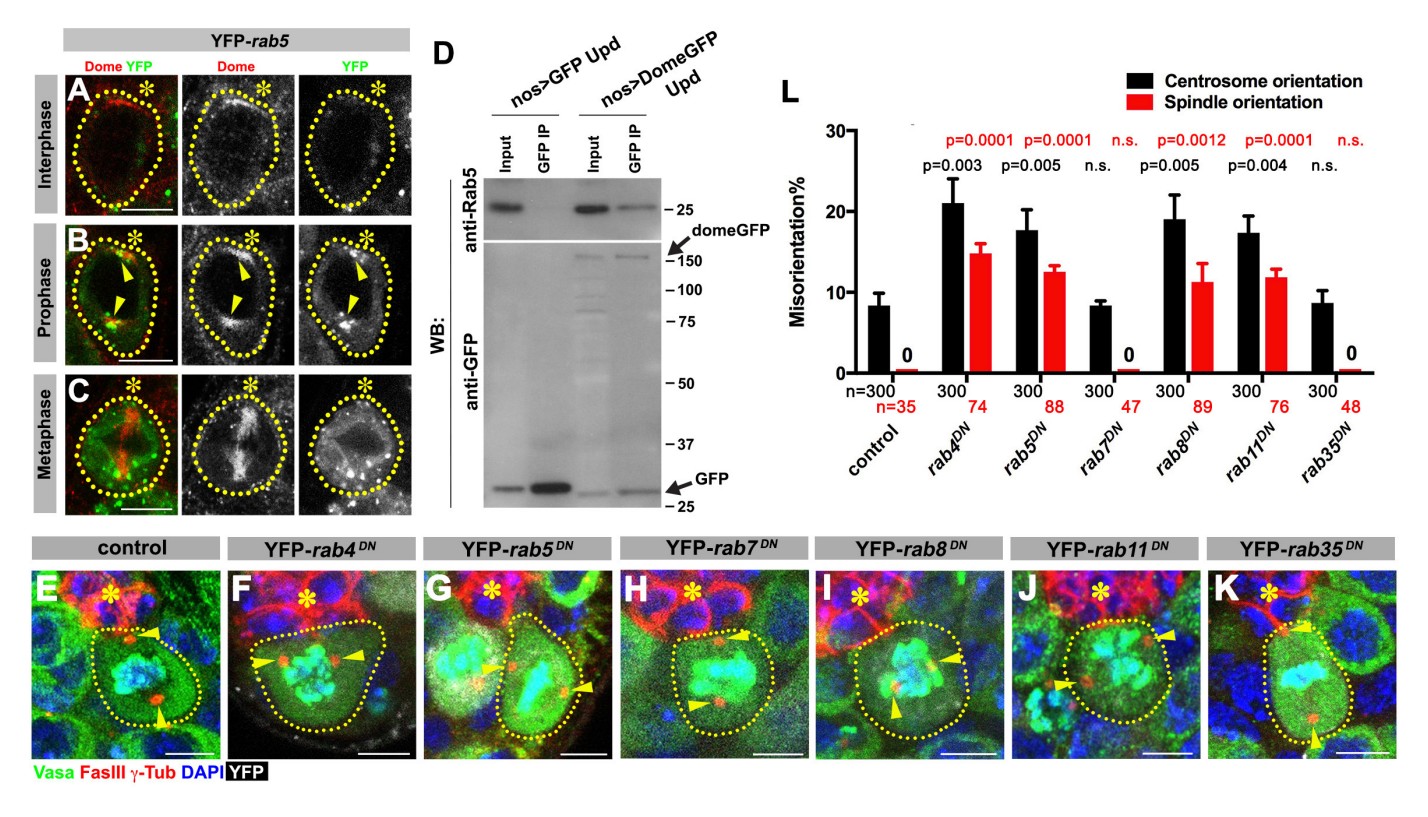

**Figure 4.** Dome is endocytosed via interactions with early/recycling endosome GTPases. (A–C) Dome localization in *nos-gal4 > UAS-YFP-rab5* GSCs at interphase (A), prophase (B), metaphase (C). Green: YFP. Red: Dome. (D) Co-immunoprecipitation of Dome with Rab5. Dome-GFP was pulled down from GSC extracts using an anti-GFP antibody and blotted using anti-Rab5 and anti-GFP antibodies. (E–K) Examples of spindles in control (E), *nos-gal4 > UAS-YFP-rab4DN* (F), *nos-gal4 > UAS-YFP-rab5DN* (G), *nos-gal4 > UAS-YFP-rab7DN* (H), *nos-gal4 > UAS-YFP-rab8DN* (I), *nos-gal4 > UAS-YFP-rab11DN* (J), *nos-gal4 > UAS-YFP-rab35DN* (K) GSCs (indicated by a white dotted line). Asterisk indicates the hub. Arrowheads indicate spindle poles. Green: Vasa and PH3. Red: Fas III and γ-Tubulin. White: YFP. Blue: DAPI. Bar: 5 µm. (L) Summary of GSC centrosome/spindle misorientation in the indicated genotypes. P value comparing control and the indicated genotype was calculated using two-tailed Student's t-test. Error bars indicate the standard deviation. N = GSC number scored for centrosome orientation or N = mitotic GSC number scored for spindle orientation.

DOI: https://doi.org/10.7554/eLife.33685.008

The following figure supplements are available for figure 4:

**Figure supplement 1.** Colocalization/immunoprecipitation of Dome with RabGTPases.
DOI: https://doi.org/10.7554/eLife.33685.009

**Figure supplement 2.** Early endosome GTPases *rab4* and *rab5* are required for correct Dome localization.
DOI: https://doi.org/10.7554/eLife.33685.010

effect (*Figure 4H,K,L*). These data suggest that internalization of Dome via endocytosis, which allows its localization with the spindle pole/spindle in mitosis, is critical for GSC centrosome/spindle orientation.

## Dome interacts with Eb1 to regulate GSC centrosome/spindle orientation

We next explored how Dome might regulate GSC centrosome/spindle orientation. By searching the *Drosophila* Interactions Database (www.droidb.org), we found that Dome is reported to interact with Eb1 (*Guruharsha et al., 2011*), a major MT-binding protein (*Akhmanova and Steinmetz, 2008*). We confirmed that endogenous Eb1 co-immunoprecipitated with Dome in *Drosophila* testes lysates enriched with GSCs (*Figure 5A*, *nos-gal4 > UAS-dome-GFP, UAS-upd*).

RNAi-mediated knockdown of *eb1* (*nos-gal4 > UAS-eb1RNAi*) dramatically increased GSC centrosome/spindle misorientation (*Figure 5B–F*), suggesting that Eb1 is involved in GSC centrosome/spindle orientation. Eb1 localized to cytoplasm during interphase and to the spindle during mitosis

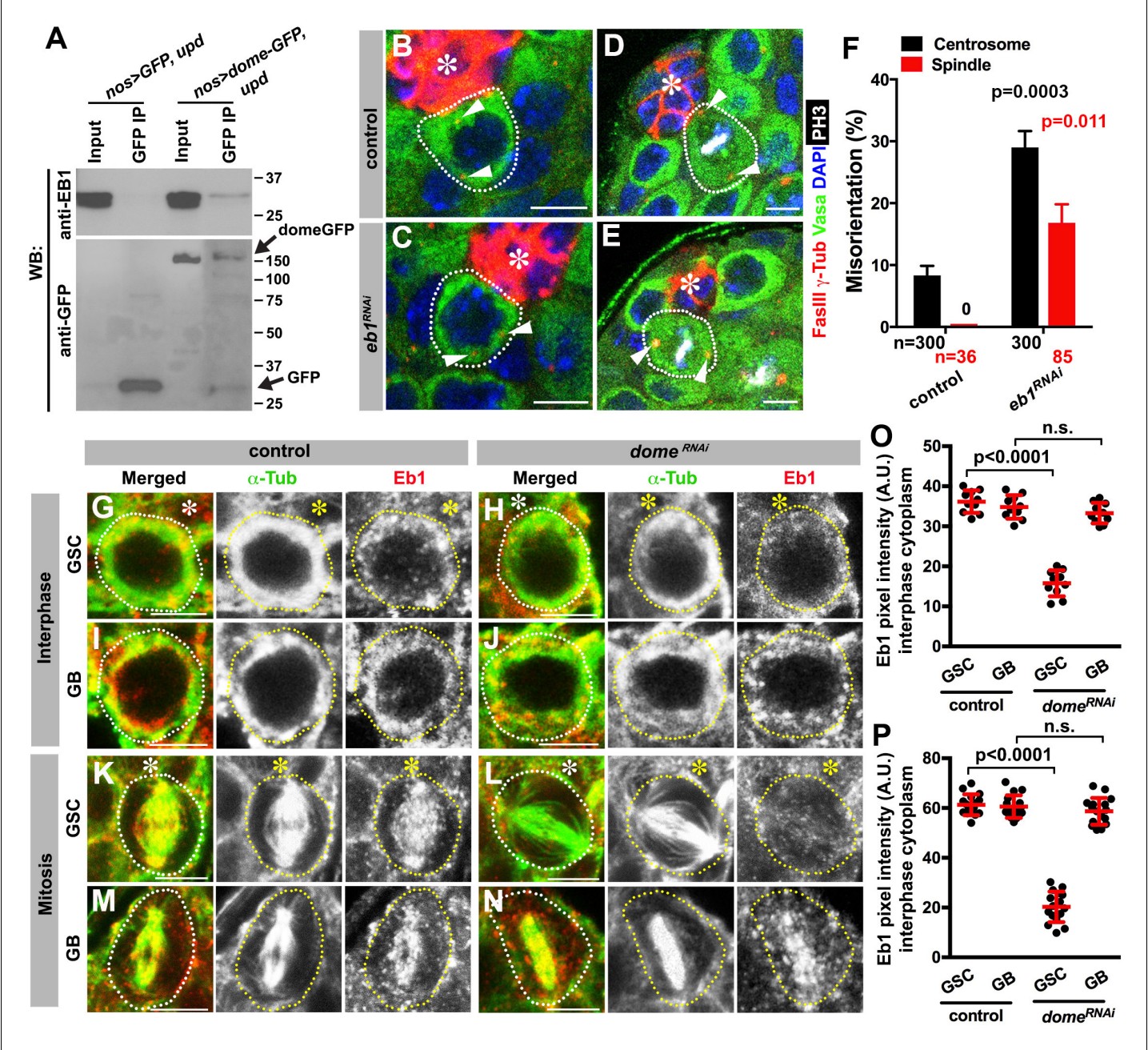

**Figure 5.** Eb1 binds to Dome and regulates GSC centrosome/spindle orientation. (**A**) Co-immunoprecipitation of Dome with Eb1. Dome-GFP was pulled down from GSC extracts using an anti-GFP antibody and blotted using anti-Eb1 and anti-GFP. (**B–C**) Examples of centrosomes in control (**B**) and *nos-gal4 > UAS-eb1$^{RNAi}$* (**C**) GSCs (outlined by a dotted line). Asterisk indicates the hub. Arrowheads indicate centrosomes. Green: Vasa. Red: Fas III and γ-Tubulin. Blue: DAPI. Bar: 5 μm. (**D–E**) Examples of mitotic spindles in control (**D**) and *nos-gal4 > UAS-eb1$^{RNAi}$* (**E**) GSCs (outlined by a dotted line). Asterisk indicates the hub. Arrowheads indicate spindle poles. Green: Vasa. Red: Fas III and γ-Tubulin. White: PH3. Blue: DAPI. Bar: 5 μm. (**F**) GSC spindle orientation in the indicated genotypes. P value was calculated using two-tailed Student's t-test. Error bars indicate the standard deviation. N = mitotic GSCs scored. (**G–J**) Eb1 localization in interphase control GSC (**G**), *nos-gal4ΔVP16, gal80$^{ts}$ > UAS-dome$^{RNAi}$* (4 d after RNAi induction) GSC (**H**), control GB (**I**), and *nos-gal4ΔVP16, gal80$^{ts}$ > UAS-dome$^{RNAi}$* (4 d after RNAi induction) GB (**J**). Asterisk indicates the hub. Green: α-Tubulin. Red: Eb1. Bar: 5 μm. (**K–N**) Eb1 localization in metaphase control GSC (**K**), *nos-gal4ΔVP16, gal80$^{ts}$ > UAS-dome$^{RNAi}$* (4 d after RNAi induction) GSC (**L**), control GB (**M**), and *nos-gal4ΔVP16, gal80$^{ts}$ > UAS dome$^{RNAi}$* (4 d after RNAi induction) GB (**N**). Asterisk indicates the hub. Green: α-Tubulin. Red: Eb1. Bar: 5 μm. (**O–P**) Pixel intensity analyses of Eb1 at interphase (**O**) and mitotic (**P**) GSCs in control and *nos-gal4ΔVP16, gal80$^{ts}$ > UAS-dome$^{RNAi}$* (4 d after RNAi induction) testes. P value was calculated using two-tailed Student's t-test. Error bars indicate the standard deviation.

DOI: https://doi.org/10.7554/eLife.33685.011

*Figure 5 continued*

The following figure supplement is available for figure 5:

**Figure supplement 1.** Eb1 is required for Dome localization in GSCs.

DOI: https://doi.org/10.7554/eLife.33685.012

in GSCs as well as in differentiating germ cells (GBs/SGs) (*Figure 5G,I,K,M*) as has been widely observed in a broad range of cell types (*Rogers et al., 2002*; *Morrison et al., 1998*). Interestingly, we found that Eb1 localization was dependent on *dome* specifically in GSCs, but not in GBs or SGs (*Figure 5H,J,L,N,O,P*), suggesting that the significance of Eb1-Dome interaction is specific to GSCs. Conversely, knockdown of *eb1* (*nos-gal4 > UAS-eb1^RNAi*) resulted in compromised localization of Dome in GSCs (*Figure 5—figure supplement 1A–D,G,H*), but not in GBs/SGs (*Figure 5—figure supplement 1E–F*), again showing that Eb1/Dome's functional interdependence is likely specific to GSCs. Taken together, these results show that Dome and Eb1 interact with each other in GSCs to regulate centrosome/spindle orientation.

## Dome interacts with Eb1 through SxIP motif in regulating GSC centrosome/spindle orientation

Eb1 is known to interact with a number of plus-end tracking proteins (+TIP) at the ends of growing MTs (*Akhmanova and Steinmetz, 2008*). Many of these +TIPs are known to contain a short, hydrophobic sequence motif (SxI/LP) through which they bind Eb1 (*Kumar and Wittmann, 2012*; *Honnappa et al., 2009*). We found this motif (SQIP) in the intracellular domain of Dome (*Figure 6A*). Indeed, we found that the SQIP sequence is essential for Dome-Eb1 interaction: Dome cytoplasmic fragment (with or without SQIP sequence) tagged with 6xHis was expressed in bacteria, purified with Ni-NTA agarose beads, and incubated with GSC extract expressing Eb1-GFP (see Materials and ethods). Whereas wild type Dome cytoplasmic fragment pulled down Eb1-GFP, Dome cytoplasmic fragment without SQIP did not (*Figure 6B*). These results show that 1) Dome cytoplasmic fragment is sufficient to interact with Eb1, and 2) SQIP sequence is critical for interacting with Eb1.

To test the functionality of SQIP sequence in vivo, we combined *dome^RNAi* with *dome-GFP* and *dome^ΔSQIP-GFP* constructs (*nos-gal4 > UAS-dome^RNAi, UAS-dome-GFP* or *nos-gal4 > UAS- dome^RNAi, UAS-dome^ΔSQIP-GFP*). These *UAS-dome* constructs were designed to be insensitive to RNAi-mediated knockdown such that their function can be tested in the absence of endogenous *dome*. Both *UAS-dome-GFP* and *UAS-dome^ΔSQIP-GFP* were fully capable of activating the JAK-STAT pathway as evidenced by upregulation of STAT in GSCs (*Figure 6C–F*), and supporting GSC self-renewal (*Figure 6G*). Despite its ability to effectively support GSC self-renewal, *dome^ΔSQIP-GFP* failed to orient centrosome/spindle (*Figure 6H–N*), demonstrating that the interaction between Dome and Eb1 via SQIP sequence is critical for GSC centrosome/spindle orientation, independent of GSC self-renewal.

To exclude the possibility that the apparent 'rescue' of *dome^RNAi* by *UAS-dome-GFP* transgenes was caused by weakened effect of *nos-gal4* when it is driving two transgenes (*UAS-dome^RNAi* and *UAS-dome-GFP*), we conducted a few additional experiments. First, adding *UAS-GFP-α-tubulin* did not reduce the severity of *dome^RNAi* (*Figure 6—figure supplement 1A–C*), suggesting that *nos-gal4* can potently drive two *UAS*-transgenes. Second, we used a 'dual *nos-gal4*' strain (containing two copies of *nos-gal4, '2xnos-gal4'*) to drive *UAS-dome^RNAi* and *UAS-dome-GFP*. These experiments (*Figure 6—figure supplement 1D,E*) recapitulated the results described above (*Figure 6G,N*), confirming that *nos-gal4*'s ability to drive multiple transgenes did not cause any complications in our interpretation of the results.

Collectively, the data shown here argue that Dome orients GSC centrosomes/spindles via its interaction with Eb1. Also, the ability of *dome^ΔSQIP-GFP* to fully rescue GSC loss due to *dome^RNAi*, while failing to rescue GSC centrosome/spindle orientation, further strengthens the notion that Dome's functions in self-renewal and centrosome/spindle orientation are separable (*Figure 6O*).

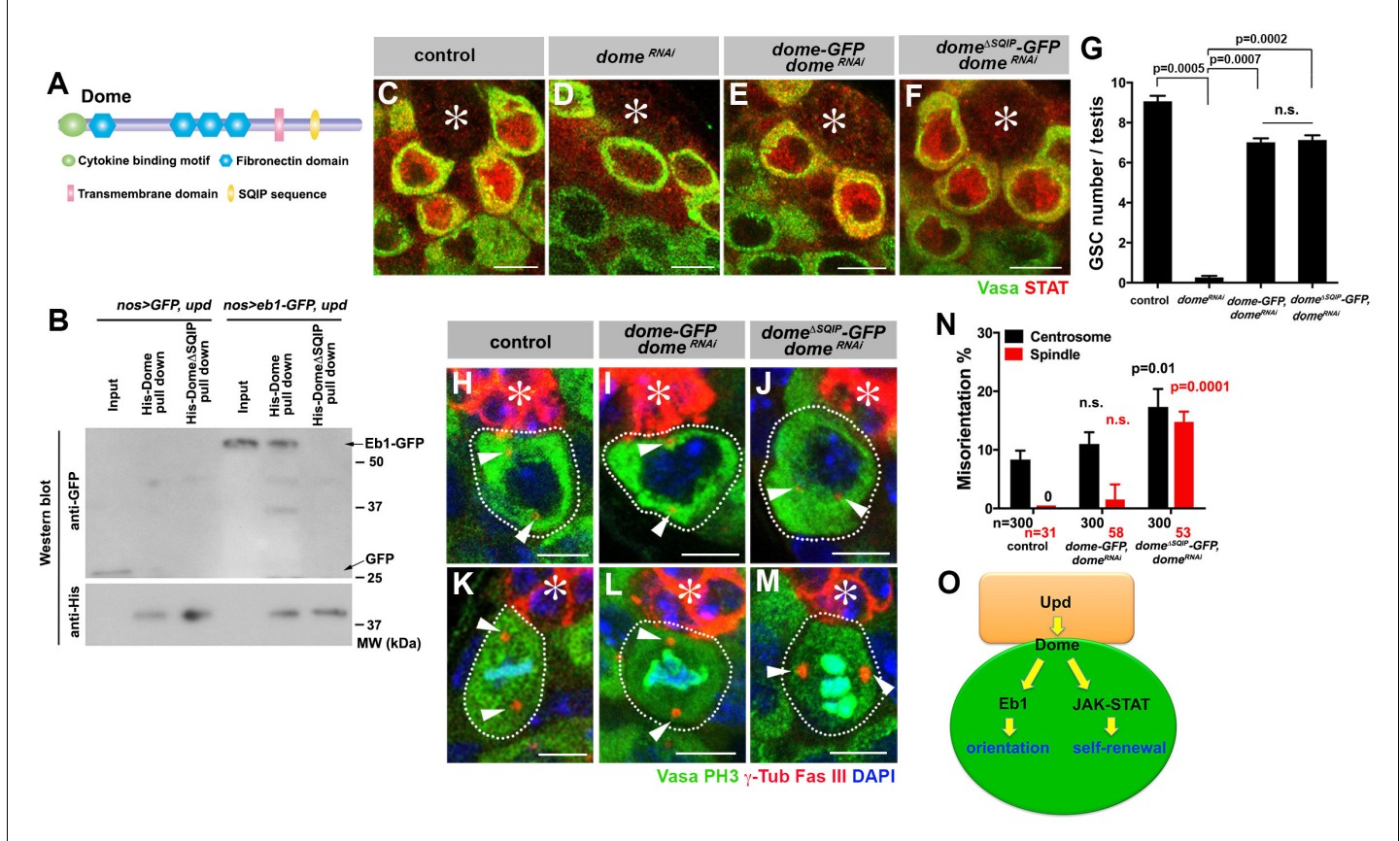

**Figure 6.** Dome interacts with EB1 via its SxIP motif to regulate spindle orientation independent of GSC self-renewal. (**A**) Domain organization of Dome. (**B**) His-tag pull-down of a 6xHis tagged cytoplasmic (C-terminal) domain of Dome (His-Dome or His-Dome$^{\Delta SQIP}$), blotted for anti-GFP and anti-His antibodies. (**C–F**) Examples of Stat92E staining in control (**C**), *nos-gal4ΔVP16, gal80$^{ts}$ > UAS-dome$^{RNAi}$* (4 d after RNAi induction) (**D**), *nos-gal4 > UAS-dome-GFP, UAS-dome$^{RNAi}$* (**E**) and *nos-gal4 > UAS-dome$^{\Delta SQIP}$-GFP, UAS-dome$^{RNAi}$* (**F**) testes. Asterisk indicates the hub. Green: Vasa. Red: Stat92E. Bar: 5 μm. (**G**) GSC numbers in control, *nos-gal4 > UAS- dome$^{RNAi}$*, *nos-gal4 > UAS-dome-GFP, UAS-dome$^{RNAi}$* and *nos-gal4 > UAS-dome$^{\Delta SQIP}$-GFP, UAS-dome$^{RNAi}$* testes. P value was calculated using two-tailed Student's t-test. Error bars indicate the standard deviation. (**H–J**) Examples of centrosmes in control (**H**), *nos-gal4 > UAS-dome-GFP, UAS-dome$^{RNAi}$* (**I**) and *nos-gal4 > UAS-dome$^{\Delta SQIP}$-GFP, UAS-dome$^{RNAi}$* (**J**) GSCs (outlined by a dotted line). Asterisk indicates the hub. Arrowheads indicate centrosomes. Green: Vasa. Red: Fas III and γ-Tubulin. Blue: DAPI. Bar: 5 μm. (**K–M**) Examples of mitotic spindles in control (**K**), *nos-gal4 > UAS-dome-GFP, UAS-dome$^{RNAi}$* (**L**) and *nos-gal4 > UAS-dome$^{\Delta SQIP}$-GFP, UAS-dome$^{RNAi}$* (**M**) GSCs (outlined by a dotted line). Asterisk indicates the hub. Arrowheads indicate spindle poles. Green: Vasa and PH3. Red: Fas III and γ-Tubulin. Blue: DAPI. Bar: 5 μm. (**N**) GSC spindle orientation in control, *nos-gal4 > UAS-dome-GFP, UAS-dome$^{RNAi}$* and *nos-gal4 > UAS-dome$^{\Delta SQIP}$-GFP, UAS-dome$^{RNAi}$* testes. P value was calculated using two-tailed Student's t-test. Error bars indicate the standard deviation. N = total GSC number scored for centrosome orientation or mitotic GSCs number scored for spindle orientation. (**O**) Model: Eb1-mediated spindle orientation and JAK-STAT-mediated self-renewal are parallel pathways downstream of Upd and Dome.

DOI: https://doi.org/10.7554/eLife.33685.013

The following figure supplements are available for figure 6:

**Figure supplement 1.** Dome$^{\Delta SQIP}$ supports GSC self-renewal but fails to orient GSC centrosome/spindle.

DOI: https://doi.org/10.7554/eLife.33685.014

**Figure supplement 2.** Localization of Dome and Eb1 is independent of E-cad, Apc1 and Apc2.

DOI: https://doi.org/10.7554/eLife.33685.015

**Figure supplement 3.** Localization of E-cad, Apc1 and Apc2 is independent of Upd-Dome-Eb1 axis.

DOI: https://doi.org/10.7554/eLife.33685.016

## Dome-Eb1 axis of centrosome/spindle orientation is mostly independent of E-cadherin or Apc1/Apc2

We previously reported that Apc1 and Apc2 regulate spindle and centrosome orientation in GSCs (*Yamashita et al., 2003*). Apc2 is recruited to the hub-GSC interface via E-cadherin to orient centrosomes (*Inaba et al., 2010*). Because Apc proteins are known to interact with Eb1 (*Su et al., 1995*;

*Nakamura et al., 2001*), we examined whether the Upd-Dome axis interacts with Apc1, Apc2 and E-cadherin. We found that Dome localization was not affected in *apc1* or *apc2* mutants (*Figure 6—figure supplement 2A–F*) or upon overexpression of dominant negative E-cadherin (*Figure 6—figure supplement 2G–H*). Conversely, localization of Eb1 was mostly unaffected under these conditions, except that Eb1 localization to the spindle was slightly compromised in *apc2* mutants (*Figure 6—figure supplement 2I–N*). Similarly, the localization of Apc1, Apc2 and E-cadherin was unaffected in *upd^{RNAi}*, *dome^{RNAi}* and *eb1^{RNAi}* testes (*Figure 6—figure supplement 3A–I*). Although we were not able to test whether these two axes function cooperatively to ensure centrosome/spindle orientation due to inability to combine all genotypes required (temporarily controlled *dome^{RNAi}* and homozygosity of *apc1* or *apc2*), above results suggest that the Upd-Dome-Eb1 and E-cad-Apc2/Apc1 axes appear to operate mostly separately (*Figure 6—figure supplement 3J*).

## Gp130 is required for orienting T cell centrosome at a model immunological synapse via interaction with Eb1 through SxI/LP motif

The above results show that a single receptor can integrate cell signaling (thus cell fate) and polarity. Such integration may play a critical role in achieving asymmetric cell divisions. Because cytokines/cytokine receptors are an evolutionarily conserved signaling module, we wondered whether our finding with Dome may hold true with other cytokine receptors. Indeed, we noticed that many cytokine receptors contain SxI/LP motifs, implying that their interaction with Eb1 may be widely conserved (*Table 1*). Gp130, the mammalian cytokine receptor with the highest homology to Dome, contains SYLP sequence in its cytoplasmic domain (*Figure 7A*). Gp130 plays a critical role in the differentiation and development of T cells via activation of the JAK-STAT signaling (*Silver and Hunter, 2010*). Immunological synapses formed between antigen-presenting cells and T-cells function as a signaling platform to facilitate T cell activation and proliferation (*Martín-Cófreces et al., 2014*). It is well-established that naïve T-cells orient their centrosomes toward the immunological synapse, leading to oriented cell division with respect to the antigen-presenting cell (*Chang et al., 2007*; *Oliaro et al., 2010*). This oriented division of naïve T-cells results in asymmetric division, generating memory and effector T cells (*Arsenio et al., 2015*).

By using the Jurkat T-cell line and anti-CD3 coated beads as a model, we examined a possible role of Gp130 in centrosome orientation at the immunological synapses. Anti-CD3 coated beads have been successfully used to activate T cells through the formation of the model immunological

**Table 1.** The list of (S/T)x(I/L)P-containing cytokine receptors.
All of these motifs are found in the cytoplasmic domain of the receptors. (S/T)x(I/L)P is underlined.

| Gene name | (S/T)x(I/L)P-containing sequence (position) |
|---|---|
| Gp130(IL6RT) | ATDEGMPKSYLPQTVRQGGY (896-915) |
| IL12RB1 | ERTEPLEKTELPEGAPELAL (627-646) |
| IL12RB2 | IAEEKTQLPLDRLL (680-693), PACPWTVLPAGDLP (782-795) |
| IL23R | PSETIPEQTLLPDEFVSCLG (580-599) |
| IL2RG | YSERLCLVSEIPPKGGALGE (325-344) |
| IL11R | SPKPGFLASVIPVDRRPGAP (401-420) |
| IL7R | LLSLGTTNSTLPPPFS LQSG (405-424) |
| IL9R | QEGPGTRLPGNLSS (376-389), GGWHLSALPGNTQS (450-463) |
| IL4R | E PPRSPQSSHLPSSSPEHLG (660-679) |
| IL3RB | GPDTTPAASDLPTEQPPSPQ (559-578) |
| EPOR | PYSNPYENSLIPAAEPLPPS (484-503) |
| GHR | PQGLILNATALPLPDKEFLS (604-623) |
| PRLR | KVNKDGALSLLPKQRENSGK (514-533) |
| TPOR | ILPKSSERTPLPLCSSQAQM (570-589) |
| CNTFR | CSWHLPTPTYIPNTFNVTVL (127-146) |

DOI: https://doi.org/10.7554/eLife.33685.018

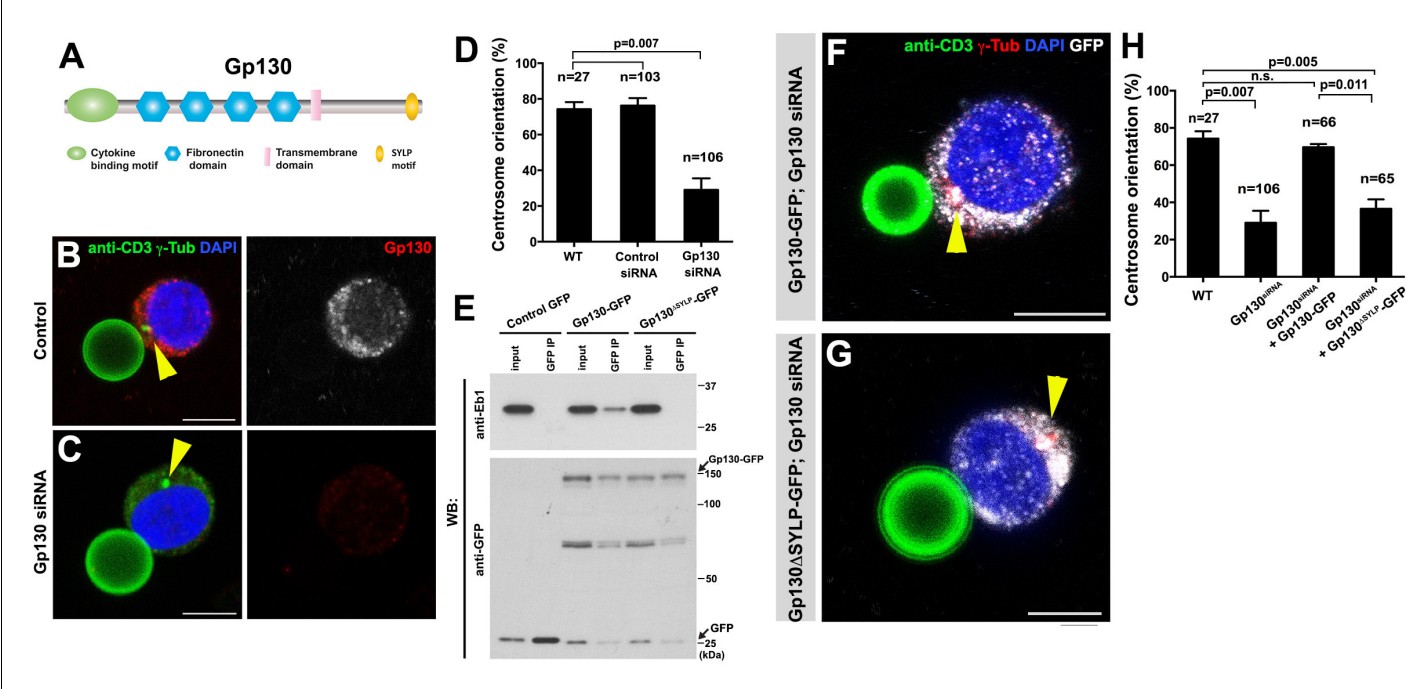

**Figure 7.** GP130 orients the centrosome at a model immunological synapse. (**A**) Domain organization of Gp130. (**B–C**) Examples of centrosome positioning (arrowheads) in control (**B**) and Gp130 siRNA (**C**) Jurkat cells. Red: Gp130. Green: anti-CD3 and γ-Tubulin. Blue: DAPI. Bar: 5 µm. (**D**) Centrosome orientation toward anti-CD3 beads in the indicated genotypes. P value was calculated using two-tailed Student's t-test. Error bars indicate the standard deviation. N = Jurkat cells scored. (**E**) Co-immunoprecipitation of Gp130 with Eb1. Control GFP, Gp130-GFP and Gp130^ΔSYLP-GFP was pulled down from Jurkat cell lysates using an anti-GFP antibody and blotted with anti-Eb1 and anti-GFP. (**F–G**) Examples of centrosome positioning (arrowheads) in Jurkat cells expressing Gp130-GFP; Gp130 siRNA (**F**) and Gp130^ΔSYLP-GFP; Gp130 siRNA (**G**). Red: γ-Tubulin. Green: anti-CD3. White, GFP. Blue: DAPI. Bar: 5 µm. (**H**) Summary of centrosome position relative to anti-CD3 beads in the indicated genotypes. P value was calculated using two-tailed Student's t-test. Error bars indicate the standard deviation. N = Jurkat cells scored.

DOI: https://doi.org/10.7554/eLife.33685.017

synapses (*Tsun et al., 2011*; *Huppa and Davis, 2003*). We confirmed that Jurkat cells orient their centrosomes toward the beads upon binding and found that Gp130 was concentrated toward the bead (*Figure 7B*). Strikingly, siRNA-mediated knockdown of Gp130 dramatically reduced centrosome orientation toward the bead (*Figure 7C–D*), suggesting that Gp130 is required for correct centrosome orientation at the immunological synapse.

Further extending the parallel between Dome and Gp130, we found that Gp130 binds Eb1 in an SYLP-dependent manner (*Figure 7E*). Moreover, the SYLP sequence is essential for centrosome orientation toward the anti-CD3-coated bead, because Gp130 lacking SYLP failed to rescue centrosome misorientation caused by siRNA-mediated knockdown of Gp130 (*Figure 7F–H*, note that rescue constructs are designed to be insensitive to siRNA). These data suggest that Eb1-binding to the cytoplasmic domain of a cytokine receptor is an evolutionarily conserved mechanism to orient the centrosomes in order to direct subsequent asymmetric cell divisions.

## Discussion

In this study, we showed the direct involvement of the niche ligand (Upd) and its receptor (Dome) in centrosome/spindle orientation in the *Drosophila* male GSCs. Critically, the role of the ligand and receptor in spindle orientation is separable from the self-renewal transcription network and mediated via direct binding of the receptor to Eb1. Although several studies revealed the involvement of signaling ligands in spindle orientation (Wnt3a ligand in division orientation of embryonic stem cells (*Habib et al., 2013*) and Wnt signal in oriented cell division in *C. elegans* embryos [*Goldstein et al., 2006*]), it remained unclear whether spindle orientation was mediated directly by the ligands or indirectly by downstream signaling cascades, such as transcriptional regulation of genes that control cell

polarity. This study provides the first example of clear separation between the signaling pathway that controls cell fate and cell polarity associated with the fate, by the use of a separation-of-function mutation (Dome$^{\Delta SQIP}$), which we found to be fully capable of supporting GSC self-renewal yet entirely compromised in orienting centrosomes/spindles. We propose that the niche ligand may provide a spatial cue to the stem cell receptor, which in turn directly regulates MTs to orient the stem cell centrosomes/spindles. Integrating activation of signaling pathway and spindle orientation into a single ligand-receptor combination has an important implication: by being responsible for both cell fate determination and orientation, the single receptor species obligatorily combines these two processes.

A few important questions arose from the present study. First, we showed that endocytosis of the receptor Dome is critical for spindle orientation. Dome and several endocytic Rab GTPases colocalized at the spindle pole during prophase, and these Rab GTPases were required for GSC centrosome/spindle orientation. It awaits future investigation whether and how endocytosis of Dome might be related to the activation of the JAK-STAT pathway: it is possible that the plasma membrane-bound form of Dome mediates JAK activation, whereas its endocytosis allows Dome to function in centrosome/spindle orientation. Alternatively, endocytosis may regulate certain aspects of signal transduction as well, as is shown in other signaling receptors, such as Notch and Egfr (*Tomas et al., 2014*; *Bray, 2016*). Curiously, when Dome localizes to the spindle during metaphase, it is not colocalized with any endocytic Rab GTPases examined (*Figure 4C*), implying that Dome might not be associated with the membrane compartment any longer. If so, it indicates that Dome is cleaved to release the intracellular domain, which contains Eb1-binding sequence (and the antigen sequence against which Dome antibody was raised).

Second, as mentioned above, defects in Upd-Dome-Eb1 axis result in misorientation of both centrosomes in interphase and spindles in mitosis, indicating that this axis is involved not only in centrosome orientation but also in the centrosome orientation checkpoint (COC). It awaits future investigation how Upd-Dome-Eb1 axis may regulate the COC, that is the sensing of centrosome misorientation. Based on the results presented in this study and considering the fact that Upd-Dome interaction would provide an ideal spatial cue for correct centrosome positioning, we speculate that Upd-Dome might contribute to dictating the location where the centrosome must be positioned, lack of which leading to defective sensing of misoriented centrosomes (i.e. defective COC activity). In our previous study, we showed that a polarity protein Bazooka forms the 'docking site' at the hub-GSC interface, to which the centrosome has to associate to satisfy the COC (*Inaba et al., 2015b*). It is tempting to speculate that Upd and/or Dome might regulate certain aspects of Baz, which in turn regulate the COC activity.

Third, the detailed mechanism of how Dome-Eb1 interaction may contribute to GSC centrosome/spindle orientation awaits future investigation. Knockdown of Dome or Eb1 did not result in spindle defects in general, as evidenced by the lack of obvious phenotypes in non-GSC germ cells (GBs/SGs). Moreover, inter-dependence of localization to the spindle between Dome and Eb1 was observed only in GSCs. These results indicate that Dome-Eb1 interaction is only relevant in GSCs. One explanation for the GSC-specific requirement of Dome-Eb1 interaction is that GSCs might have distinct microtubule characteristics, for example dynamics and/or asymmetry. In contrast to symmetrically dividing cells (including non-GSC germ cells), GSCs are in the need of orienting spindles along their polarity axis, which likely requires microtubule/spindle pulling and/or anchoring. Dome might modulate microtubule characteristics by recruiting Eb1 to the microtubule. Although Eb1 can bind to and regulate microtubule behavior on its own, GSC-specific microtubule characteristics might necessitate Dome's aid for Eb1 to function properly.

Our study further revealed a striking conservation of the cytokine receptor-Eb1 axis to orient the centrosomes in diverse systems: as distant as the *Drosophila* male GSCs and mammalian (model) immunological synapse. This mechanism may be widely utilized by signaling cells (e.g. niche cells and antigen-presenting cells) to ensure asymmetric cell division of their target cells (e.g. stem cells and T cells). Considering the conservation of SxI/LP motif in many other cytokine receptors, it warrants future investigation to study their potential roles in regulation of cell polarity and/or microtubule dynamics via binding to Eb1.

In summary, our study demonstrates that a single receptor regulates cell fate and cell polarity, thereby coordinating two critical aspects of asymmetric cell division.

## Materials and methods

### Fly husbandry, strains and transgenic flies

All fly stocks were raised on standard Bloomington medium at 25°C, and young flies (0- to 1-day-old adults) were used for all experiments unless otherwise noted. The following fly stocks were used: *nos-gal4* (*Van Doren et al., 1998*), *UAS-upd* (*Zeidler et al., 1999*), *tub-gal80^ts* (*McguireMcGuire et al., 2003*), *UAS-dome-EGFP* (*Ghiglione et al., 2002*), *apc2^d40*/TM3, *apc2^ΔS*/ TM6b, *apc1^Q8* and *Df(3R)3450* (*Yamashita et al., 2003*) (obtained from Bloomington Drosophila Stock Center (BDSC)), *hop^tum-1* (*Corwin and Hanratty, 1976*), *stat92E^06346* (*Hou et al., 1996*), *stat92E^F* (*Baksa et al., 2002*), *UAS-upd^RNAi* (P{GD1158}v3282 from Vienna *Drosophila* RNAi Center), *UAS-dome^RNAi* (TRiP.HMS00647, TRiP.HMS01293, BDSC), *UAS-hop^RNAi* (TRiP.HMS00761, BDSC), *UAS-stat92E^RNAi* (TRiP.HMS00035, BDSC), *UAS-eb1^RNAi* (TRiP.GL00559, BDSC), *nos-gal4* without VP16 (*nos-gal4ΔVP16*) (*Inaba et al., 2015a*), *hs-FLP, nos > stop > gal4 UAS-eGFP/CyO* (*Salzmann et al., 2013*). *hs-FLP, UAS-GFP Act-FRT-stop-FRT-gal4, UAS-DEFL, UAS-dCR4h* and *UAS-GFP-Apc2* (*Inaba et al., 2010*), *UAS-RFP-Apc1* (*Mattie et al., 2010*), *upd-gal4* (Flybase; http://fly-base.org/reports/FBti0002638.html). Transgenic flies carrying wild-type and dominant negative (DN) variants of *UAS-YFP-rab4, UAS-YFP-rab5, UAS-YFP-rab7, UAS-YFP-rab8, UAS-YFP-rab11,* and *UAS-YFP-rab35* were obtained from Bloomington Stock Center and have been described previously (*Zhang et al., 2007*).

For the construction of *UAS-dome^ΔSQIP-GFP*, a plasmid encoding pMAT-*dome^ΔSQIP* (insensitive to RNAi) was generated by gene synthesis (Life Technologies). The *dome^ΔSQIP* fragment was sequenced for validation and subcloned into EcoRI/XhoI sites of *pUAST-EGFP-attB*. *UAS-dome-GFP* (insensitive to RNAi) was generated by site-specific insertion of SQIP sequence from p*UAST-dome^ΔSQIP-GFP* plasmid using the following primers: 5'-gtcccagccgctgtcccagattccgctcagcggctacg-3' and 5'-cgtagccgctgagcggaatctgggacagcggctgggac-3'. Transgenic flies were generated using PhiC31 integrase-mediated transgenesis at the P{CarryP}attP2 (FBst0008622) integration site (BestGene).

For construction of 6xHis-tagged Dome cytoplasmic domain, Dome cytoplasmic domain was amplified by PCR from full-length cDNA (LD46805) (Drosophila Genomics Resource Center) using the following primers: 5'-gtagaattcggtctagtgctgccgcag-3' and 5'-gtactcgagttagaggacgtgccgattgtg-3'. The PCR product was cloned into EcoRI/XhoI sites of pET28a with an N-terminal 6xHis tag, yielding pET28a-6xHis-DomeC plasmid. The *dome* cytoplasmic fragment without the SQIP sequence was generated by site-specific deletion of SQIP sequence from pET28a-6xHis-DomeC plasmid using the following primers: 5'-ctcagcggctacgtgcc-3' and 5'-cagcggctgggacatcg-3'.

### Cell culture, siRNA transfection, and retrovirus infection of Jurkat cells with Gp130-GFP constructs

A human T cell line Jurkat cells (obtained from Malini Raghavan, University of Michigan, authenticated by ATCC and determined to be mycoplasma-negative prior to use) were grown in RPMI1640 media (Life Technologies, Carlsbad, CA) supplemented with 2 μM L-glutamine in a 37°C humidified incubator at 5% $CO_2$. BOSC cells (obtained from Malini Raghavan, University of Michigan, authenticated by ATCC and determined to be mycoplasma-negative prior to use) were grown in DMEM media (Life Technologies). Culture media were supplemented with 10% fetal calf serum, and 100 U/mL penicillin and streptomycin (Life Technologies). siRNA transfections were performed using electroporation as described previously (*Jordan et al., 2008*). For all siRNA experiments, the siRNAs were transfected at a final concentration of 20 nM. Seventy-two hours after initial transfection, cells were harvested and used for experiments.

For generation of Gp130 constructs, the coding sequence of Gp130 (Dharmacon) without stop codon was amplified with PCR using the primers 5'-gtaaagcttatgttgacgttgcagacttggg-3' and 5'-gtaggatccctgaggcatgtagccgcc-3', and cloned into the pcDNA3.1-GFP vector. Following confirmation by DNA sequencing, the GFP fusion protein construct was cloned into a mouse stem cell virus (MSCV) retroviral expression vector (a generous gift of Malini Raghavan, University of Michigan). MSCV-Gp130^ΔSYLP-GFP was generated by site-specific deletion of SYLP sequence from MSCV-Gp130-GFP plasmid using the following primers: 5'-ctgatgaaggcatgcctaaacagactgtacggc-3 and 5'-gccgtacagtctgtttaggcatgccttcatcag-3'.

Retrovirus was produced by transient transfection of BOSC cells (*Rizvi et al., 2014*). The Jurkat cells were infected with retroviruses encoding GFP, Gp130-GFP or Gp130$^{\Delta SQIP}$-GFP. Then, 1 mg/mL G418 (Life Technologies) was added for drug selection to establish stable cell lines and maintained with 0.5 mg/mL G418.

## Immunofluorescent staining and confocal microscopy

*Drosophila* testes were dissected in phosphate-buffered saline (PBS), transferred to 4% formaldehyde in PBS and fixed for 30 min. The testes were then washed in PBST (PBS containing 0.1% Triton X-100) for at least 30 min, followed by incubation with primary antibody in 3% bovine serum albumin (BSA) in PBST at 4°C overnight. Samples were washed for 60 min (three 20 min washes) in PBST, incubated with secondary antibody in 3% BSA in PBST at 4°C overnight, washed as above, and mounted in VECTASHIELD with 4′,6-diamidino-2-phenylindole (DAPI; Vector Labs, Burlingame, CA). For Eb1 staining, *Drosophila* testes were fixed in 4% formaldehyde in methanol (pre-chilled at −20°C) for 10 min. The testes were then washed in PBST for 60 min and blocked in 3% bovine serum albumin (BSA) for 60 min, followed by antibody staining and mounting as described above.

To examine centrosome reorientation in Jurkat cells, the previously published protocol was followed (*Tsun et al., 2011*). In brief, streptavidin-conjugated beads (Polysciences, Warrington, PA) were diluted and washed three times 1%BSA/PBS buffer. $5 \times 10^7$ beads/mL were coated with 10 μg/mL biotinylated anti-CD3 antibody (Biolegend, San Diego, CA) for 1 hr at 4°C, then washed three times and resuspended in 1%BSA/PBS buffer at the concentration of $5 \times 10^6$ beads/mL. 20 μL of anti-CD3 antibody-coated beads were seeded onto poly-L-lysine-treated slide chamber in serum-free RPMI medium at the concentration of $10^6$ beads/mL for 1 hr. Then 200 μL Jurkat cells ($10^5$ cells/mL) were added and incubated for 30 min at 37°C to allow conjugation before fixation. Cell-bead conjugates were washed with PBS, fixed in pre-chilled methanol for 5 min, and washed extensively in PBS and blocked with PBS with 3% BSA for 2 hr. All antibodies were diluted in 3% BSA in PBS. The samples were stained by incubating with a primary antibody for 3 hr, followed by a secondary antibody incubation for 1 hr. Extensive washing with PBS was performed in between each step. Samples were mounted in VECTASHIELD with DAPI.

The primary antibodies used were as follows: mouse anti-Fasciclin III [1:20; developed by C. Goodman, obtained from Developmental Studies Hybridoma Bank (DSHB)]; mouse anti-α-Tubulin (4.3; 1:100; developed by C. Walsh, obtained from DSHB); rat anti-Vasa (1:20; developed by A. Spradling and D. Williams, obtained from DSHB); mouse anti-γ-Tubulin (GTU-88; 1:100; Sigma-Aldrich, St. Louis, MO); rabbit anti-Ser10-phosphorylated Histone H3 (1:200; Millipore, Burlington, MA); rabbit anti-Vasa (d-26; 1:200; Santa Cruz Biotechnology, Santa Cruz, CA); rabbit anti-Eb1 (1: 200; a gift from Stephen Rogers) (*Rogers et al., 2002*); rabbit anti-Gp130 (1:500, Santa Cruz Biotechnology); rabbit anti-γ-tubulin (1:500; abcam); chicken anti-GFP (1:1000, Aves Labs, Tigard, OR); guinea pig anti-Stat92E (1:100) (*Inaba et al., 2015a*). Anti-Dome antibody was generated by injecting a peptide (CVDRDGYDDNHETGPISA) into rabbits (Covance, Denver, PA). Specificity of the serum was validated by the lack of staining in *dome*$^{RNAi}$ testis. Alexa Fluor-conjugated secondary antibodies (Life Technologies) were used with a dilution of 1:200. Images were taken using an upright Leica TCS SP8 confocal microscope with a 63 × oil immersion objective (NA = 1.4) and processed using Adobe Photoshop software.

Correct centrosome orientation was defined as at least one centrosome being closely associated with the hub-GSC junction or the bead-Jurkat cell interface. Correct spindle orientation was defined as one spindle pole being juxtaposed to the hub-GSC junction in mitosis. Conversely, misoriented centrosomes/spindles are defined as neither of centrosomes/spindle poles being near the hub-GSC junction. Based on these criteria, orientation was scored as binary outcomes (oriented or misoriented).

## Co-immunoprecipitation, His-tag pull down and western blotting

For immunoprecipitation using *Drosophila* testis lysate, testes enriched with GSCs due to ectopic Upd expression (100 pairs/sample) were dissected into PBS at room temperature within 30 min. Testes were then homogenized and solubilized with lysis buffer (10 mM Tris-HCl pH 7.5; 150 mM NaCl; 0.5 mM EDTA supplemented with 0.5% NP40 and protease inhibitor cocktail (EDTA-free, Roche, Switzerland)) for 30 min at 4°C. For immunoprecipitation using Jurkat cell lysates, cell lysates

($2 \times 10^7$ cells/sample) were prepared in PBS with 2% Trition X-100, 150 mM NaCl, 50 mM Tris-HCl, pH 8.0, 1 mM $MgCl_2$ and protease inhibitor cocktail. Testes or Jurkat cell lysates were centrifuged at 13,000 rpm for 15 min at 4°C using a table centrifuge, and the supernatants were incubated with GFP-Trap magnetic agarose beads (ChromoTek, Germany) for 4 hr at 4°C. The beads were washed three times with wash buffer (10 mM Tris-HCl pH 7.5; 150 mM NaCl; 0.5 mM EDTA). Bound proteins were resolved in SDS-PAGE and analyzed by western blotting.

For His-tag pull-down assays, 6xHis-tagged Dome fragments (with or without SQIP) were expressed in *E. coli*. The cultures were pelleted, and resuspended in lysis buffer (50 mM phosphate buffer, pH 8.0; 300 mM NaCl, 10 mM imidazole, 0.1% NP40 and protease inhibitor cocktail (EDTA-free, Roche)) for 30 min at 4°C. After centrifugation at 13,000 rpm for 15 min at 4°C, supernatants were incubated with Ni-NTA agarose beads for 1 hr at 4°C. The beads were washed twice with wash buffer (50 mM phosphate buffer, pH 8.0; 300 mM NaCl, 20 mM imidazole). Then, testes extracts expressing Eb1-GFP or GFP enriched with GSCs (*nos-gal4 > UAS-upd, UAS-eb1-GFP* or *nos-gal4 > UAS-upd, UAS-GFP*) were added to the beads and incubated for 2 hr at 4°C. The beads were washed three times with wash buffer. The bound proteins were eluted with elution buffer (50 mM phosphate buffer, pH 8.0; 300 mM NaCl, 250 mM imidazole) and analyzed by SDS-PAGE followed by western blotting using anti-His and anti-GFP antibodies.

For western blotting, samples subjected to SDS-PAGE (NuPAGE Bis-Tris gels (8%; Invitrogen, Carlsbad, CA)) were transferred onto polyvinylidene fluoride (PVDF) membranes (Immobilon-P; Millipore). Membranes were blocked in PBS containing 5% nonfat milk and 0.1% Tween-20, followed by incubation with primary antibodies diluted in PBS containing 5% nonfat milk and 0.1% Tween-20. Membranes were washed with PBS containing 5% nonfat milk and 0.1% Tween-20, followed by incubation with secondary antibody. After washing with PBS, detection was performed using an enhanced chemiluminescence system (Amersham, UK). Primary antibodies used were rabbit anti-GFP (abcam; 1:4000), mouse anti-His tag (Pierce; 1:2000), rabbit anti-Rab4 (a gift from Victor Hatini; 1:2000) (*de Madrid et al., 2015*), rabbit anti-Rab7 (a gift from Akira Nakamura; 1:4000) (*Tanaka and Nakamura, 2008*), rabbit anti-Rab11 (a gift from Akira Nakamura; 1:4000), rabbit anti-Rab5 (abcam; 1:4000), mouse anti-Rab8 (BD Biosciences; 1:4000), rabbit anti-Eb1 (a gift from Stephen Rogers; 1:4000), mouse anti-Eb1 (Santa Cruz Biotechnology; 1:1000). Secondary antibodies used were goat anti-mouse IgG and goat anti-rabbit IgG conjugated with horseradish peroxidase (HRP) (abcam; 1:5000).

## Data analyses

Statistical analysis was performed using GraphPad Prism seven software. For centrosome and spindle orientation scoring, 300 GSCs were scored for centrosome misorientation, and >25 mitotic GSCs were scored for spindle misorientation. Centrosome misorientation referred to no close association of neither of the two centrosomes with the hub-GSC inteface during interphase. Spindle misorientation referred to no close association of neither of the two spindle poles with the hub-GSC inteface during mitosis. For GSC number quantification, 300 GSCs were scored. Data are shown as means ± standard deviation. The *P*-value (two-tailed Student's *t-test*) is provided for comparison with the control. Pixel intensity analysis was performed by manually drawing regions of interest in ImageJ and background intensity was subtracted from each value.

## Acknowledgements

We thank Drs. Stephen Rogers, Malini Raghavan, Melissa Rolls, Victor Hatini, Akira Nakamura, the Bloomington Stock Center, the Vienna Drosophila RNAi Center, Drosophila Genomics Resource Center and the Developmental Studies Hybridoma Bank for reagents, and the Yamashita laboratory for discussions and comments on the manuscript. This work was supported by R01GM07200606 (to AJH, MM and YMY) and R01GM118308 (to YMY). The research in the Yamashita laboratory is supported by Howard Hughes Medical Institute.

# Additional information

### Competing interests

Yukiko M Yamashita: Reviewing editor, *eLife*. The other authors declare that no competing interests exist.

### Funding

| Funder | Grant reference number | Author |
|---|---|---|
| Howard Hughes Medical Institute | | Yukiko M Yamashita |
| National Institute of General Medical Sciences | R01GM07200606 | Alan J Hunt<br>Michael Mayer<br>Yukiko M Yamashita |
| National Institute of General Medical Sciences | R01GM118308 | Yukiko M Yamashita |

The funders had no role in study design, data collection and interpretation, or the decision to submit the work for publication.

### Author contributions

Cuie Chen, Conceptualization, Formal analysis, Validation, Investigation, Methodology, Writing—original draft, Writing—review and editing; Ryan Cummings, Investigation; Aghapi Mordovanakis, Methodology; Alan J Hunt, Michael Mayer, David Sept, Supervision, Methodology; Yukiko M Yamashita, Conceptualization, Supervision, Investigation, Writing—original draft, Project administration, Writing—review and editing

### Author ORCIDs

Cuie Chen (iD) http://orcid.org/0000-0002-5498-9753
Ryan Cummings (iD) http://orcid.org/0000-0003-0540-9174
Yukiko M Yamashita (iD) http://orcid.org/0000-0001-5541-0216

### Decision letter and Author response

Decision letter https://doi.org/10.7554/eLife.33685.021
Author response https://doi.org/10.7554/eLife.33685.022

# Additional files

### Supplementary files

• Transparent reporting form
DOI: https://doi.org/10.7554/eLife.33685.019

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
