## [Decision Letter]

Thank you for submitting your article "Cytokine receptor-Eb1 interaction couples cell polarity and fate during asymmetric cell division" for consideration by *eLife*. Your article has been reviewed by two peer reviewers, and the evaluation has been overseen by K VijayRaghavan as the Senior and Revieiwing Editor. The following individuals involved in review of your submission have agreed to reveal their identity: Sonia Sen and Terry Lechler.

The reviewers have discussed the reviews with one another and the Reviewing Editor has drafted this decision to help you prepare a revised submission.

Summary:

This manuscript identified an unexpected pathway required to orient the mitotic spindle in *Drosophila* male germline stem cells that appears to be evolutionarily conserved. The Unpaired/Dome ligand/receptor are required for two distinct aspects of germ cell biology. They had a previously defined role in maintaining germline stem cells. Here, a separable function in orienting the spindle is demonstrated. Dome, therefore, joins Wnt receptors, GPCR's, and E-cadherin as transmembrane receptors that can control underlying spindle orientation. Further, the manuscript demonstrates that Dome's spindle orienting machinery requires an interaction with the microtubule binding protein Eb1. Notably, many cytokine receptors have a conserved Eb1-binding motif and this appears to be important for centrosome localization at the immunological synapse of T cells. This work will be of broad interest to those studying germ cell/stem cell development, cytokine receptor signaling and asymmetric cell divisions.

Although these are very interesting findings, there are few issues that we have listed below, which need to be addressed to support the claims that are being made.

Essential revisions:

1) We were confused by the cell cycle timing of requirements for Dome/Rabs/Eb1 function. If the centrosome position is altered in interphase, then it suggests that at least some of the action is occurring before mitosis. However, it is not clear whether Dome is internalized (or needs to be internalized in interphase). Can the authors more precisely define when (or where) Dome is required (membrane, centrosomes, spindle microtubules?)

2) Upd regulating Dome localization is an important piece of the story because it implies that activation of the JAK/STAT pathway is necessary for the translocation of Dome to the spindle. (It is likely also what positions one centrosome near the hub.) Given the importance of this, we find the data supporting it not entirely convincing. While the knock-down of Upd shows a clear lack of enrichment of Dome at the hub-GSC interface at interphase, during mitosis, the localization appears to be normal, though maybe fainter. Maybe the authors could quantify this observation to support their claim?

3) Furthermore, the authors have generated clones that over-express Upd to test if this can drive the localization of Dome in adjacent cells. This would be an excellent support for their claim, however, it appears from the image that they are quantifying the localization of Dome within the clone, and not in the cells adjacent to the clone. (Figure 3G is not convincing as the picture provided looks very similar to 3E). Centrosome staining to demonstrate that this ectopic localization is not centrosome-associated would strengthen the finding. Could the authors please clarify the details of this experiment?

4) The authors show that Dome co-localises with early, but not late or recycling endosomal markers and that dominant negative forms of the relevant Rabs causes orientation defects similar to Upd or Dome loss-of-function. While these independent, yet similar effects of Dome LOF and Rab LOF on centrosome/spindle suggest they act together, it would be nice to see a link between the abrogated Rab compartments and localization of Dome. Does over-expression of the relevant DN Rabs result in a lack of Dome localization to the spindle? (We appreciate the dangers of interpreting DN over-expression, so these experiments need to be calibrated carefully and interpreted with care. These are not 'required' experiments.)

5) The deltaSQIP section is extremely interesting because it very nicely delineates the transcriptional and non-transcriptional roles of JAK/STAT signaling in GSC identity/spindle orientation. However, we are a little concerned about the controls used. If we understand correctly, the authors knock down Dome with RNAi, and in subsequent panels, drive rescue constructs (UAS-*dome-GFP* or UAS-Dome-deltaSQIP-GFP) with the same Gal4. While it’s true that one construct shows a more significant rescue than the other, we are concerned that the level of knockdown of RNAi is insufficient while two UAS constructs are being driven by the same Gal4. This might result in a more attenuated phenotype that could be mistaken for a rescue. Given the importance of this finding, we would urge the authors to please use the appropriate controls in this experiment.

6) Please comment on the severity of the mis-orientation phenotype relative to the E-cad/APC phenotypes and address whether the defects are additive or not.

7) Please provide a quantification of the defects in Eb1/Dome localization when the other is perturbed (Figure 5 and Figure 5—figure supplement 1).

8) In some of their single-channel images they've used grey-scale, and in others, a LUT. We recommend using grey-scale for all.

---

## [Author Response]

1) We were confused by the cell cycle timing of requirements for Dome/Rabs/Eb1 function. If the centrosome position is altered in interphase, then it suggests that at least some of the action is occurring before mitosis. However, it is not clear whether Dome is internalized (or needs to be internalized in interphase). Can the authors more precisely define when (or where) Dome is required (membrane, centrosomes, spindle microtubules?)

This is an important point, and we appreciate the reviewers for pointing this out. The newly added data (Figure 4—figure supplement 2) indicate that interphase dome localization is also perturbed upon expression of dominant negative Rab4 or Rab5. This suggests that even interphase dome localization requires endocytosis, which explains their requirement in centrosome orientation. Whether interphase Dome, observed near the hug-GSC junction, is already internalized or not is somewhat difficult with the resolution of light microscopy.

Based on these data, the most likely interpretation is that Dome is constantly required for centrosome or spindle orientation wherever it is localized. However, formal testing of the idea would require the generation of Dome mutant protein that can localize to the membrane but cannot be endocytosed, one that cannot translocate to the spindle, etc., which would require a much deeper knowledge of exactly how Dome may be endocytosed.

2) Upd regulating Dome localization is an important piece of the story because it implies that activation of the JAK/STAT pathway is necessary for the translocation of Dome to the spindle. (It is likely also what positions one centrosome near the hub.) Given the importance of this, we find the data supporting it not entirely convincing. While the knock-down of Upd shows a clear lack of enrichment of Dome at the hub-GSC interface at interphase, during mitosis, the localization appears to be normal, though maybe fainter. Maybe the authors could quantify this observation to support their claim?

Now we include pixel intensity analysis which clearly shows down regulation of Dome under upd-RNAi condition (new Figure 3G, H). Although pixel intensity analysis has a caveat of possible tissue-to-tissue variation in staining, we are fairly confident that our quantification reflects real reduction in Dome in GSCs upon upd knockdown, because upd-RNAi did not cause any reduction in Dome level in GBs/SGs, and Dome level in GBs/SGs provided a nice control to show that the staining variation did not cause issues in interpretation.

We believe that our data show that Upd’s binding to Dome is important for Dome’s localization. However, it does not necessarily mean that JAK-STAT activation is necessary for Dome localization: indeed we did not see Dome’s localization problem upon knockdown of JAK or STAT (now added as Figure 3—figure supplement 2), showing that JAK or STAT is not required for Dome localization.

3) Furthermore, the authors have generated clones that over-express Upd to test if this can drive the localization of Dome in adjacent cells. This would be an excellent support for their claim, however, it appears from the image that they are quantifying the localization of Dome within the clone, and not in the cells adjacent to the clone. (Figure 3G is not convincing as the picture provided looks very similar to 3E). Centrosome staining to demonstrate that this ectopic localization is not centrosome-associated would strengthen the finding. Could the authors please clarify the details of this experiment?

Figure 3G (now Figure 3I in the revised version) shows the case of a somatic cell (cyst stem cell) being a Upd-expressing clone, which encapsulates a GSC. Because cyst stem cells adopt a thin/squamous shape, it may look as if it is a cytoplasm of GSC. But based on cellular markers etc. we can confidently say that Figure 3G (now 3I) shows a GSC encapsulated by a Upd-expressing cyst stem cell. We have added a cartoon picture to this figure panel to explain which cells are which.

We agree that the previous Figure 3B (we think the reviewer meant Figure 3B, by ‘3E’) and 3G (now 3I) looks very similar. However, 3B represents prophase, when Dome accumulates to the spindle poles, whereas 3G (now 3I) represents interphase GSCs, when Dome should only be at the hub-GSC interphase. We agree that the former 3B very much looked like Figure 3G (now 3I) because of somewhat stretched appearance of Dome pattern, although we typically see rounder Dome on the spindle pole in prophase. Therefore, to avoid the confusion, we replaced 3B with a new representative image.

4) The authors show that Dome co-localises with early, but not late or recycling endosomal markers and that dominant negative forms of the relevant Rabs causes orientation defects similar to Upd or Dome loss-of-function. While these independent, yet similar effects of Dome LOF and Rab LOF on centrosome/spindle suggest they act together, it would be nice to see a link between the abrogated Rab compartments and localization of Dome. Does over-expression of the relevant DN Rabs result in a lack of Dome localization to the spindle? (We appreciate the dangers of interpreting DN over-expression, so these experiments need to be calibrated carefully and interpreted with care. These are not 'required' experiments.)

Now we include the data to show dominant negative forms of early endosomal GTPases Rab4 and Rab5 (Rab4^DN^ and Rab5^DN^) causes Dome localization defects (Figure 4—figure supplement 2).

5) The δ SQIP section is extremely interesting because it very nicely delineates the transcriptional and non-transcriptional roles of JAK/STAT signaling in GSC identity/spindle orientation. However, we are a little concerned about the controls used. If we understand correctly, the authors knock down Dome with RNAi, and in subsequent panels, drive rescue constructs (UAS-Dome-GFP or UAS-Dome-deltaSQIP-GFP) with the same Gal4. While it’s true that one construct shows a more significant rescue than the other, we are concerned that the level of knockdown of RNAi is insufficient while two UAS constructs are being driven by the same Gal4. This might result in a more attenuated phenotype that could be mistaken for a rescue. Given the importance of this finding, we would urge the authors to please use the appropriate controls in this experiment.

As pointed out by the reviewers, whether adding multiple UAS-transgenes (UAS-Dome-GFP and UAS-Dome-RNAi) might weaken the expression of each transgene, leading to ‘apparent’ rescue is a critically important issue. If it were the case, the validity of our central conclusion would be in question. We did not think it would be the case (see below for the reasoning), but we also added a few more experiments to further validate this notion.

a) Originally, we reasoned that, if the level of knockdown becomes insufficient by adding another UAS-transgene, we would not have observed a severe spindle misorientation with dome-RNAi + dome-DSQIP (Figure 6N). Thus, we did not further test the possibility raised by the reviewers (but we agree this point should be addressed at the highest rigor, given its importance in this paper).

b) To further test whether addition of UAS-transgene(s) might reduce the expression level of individual transgenes, resulting in reduced knockdown efficiency, we added UAS-GFP-a-tubulin to the UAS-Dome-RNAi construct (new Figure 6—figure supplement 1A-C). If having two UAS-transgenes reduces knockdown efficiency, we would expect to see less severe germ cell loss phenotype. As is now shown in Figure 6—figure supplement 1A-C, nos>UAS-DomeRNAi, UAS-GFP-tubulin showed an equally severe GSC loss when compared to nos>UAS-DomeRNAi, suggesting that addition of UAS-GFP-a-tubulin did not reduce the efficacy of UAS-DomeRNAi.

c) To further protect against ‘weakened efficacy’ of nos-gal4 driver when overloaded by multiple UAS-transgenes, we used ‘dual nos-gal4’ strains (two nos-gal4 drivers on 2^nd^ and 3^rd^ chromosomes). These experiments (Figure 6—figure supplement 1D, E) also recapitulated the data as shown in Figure 6G.

6) Please comment on the severity of the mis-orientation phenotype relative to the E-cad/APC phenotypes and address whether the defects are additive or not.

Unfortunately, we have not been able to address this point. Temporarily controlled dome-RNAi requires manipulation of 3 chromosomes (nos-gal4, gal80ts, UAS-domeRNAi), and adding apc1 or apc2 homozygous mutation requires 2 chromosomes (out of 5 chromosomes that can be used for genetics in diploid organisms (X and 2x2^nd^, 2x3^rd^)), and some genes/insertions required to achieve this genetic combination map very close to each other of the same chromosome. Although this is a very interesting question to ask, we are still working to overcome this technical difficulty.

Therefore, we added the comment in the text to explain this caveat that the functional relationship (whether the effect may be additive or not) is not fully tested.

7) Please provide a quantification of the defects in Eb1/Dome localization when the other is perturbed (Figure 5 and Figure 5—figure supplement 1).

Now we provide the pixel intensity analysis in Figure 5O, P and Figure 5—figure supplement 1.

8) In some of their single-channel images they've used grey-scale, and in others, a LUT. We recommend using grey-scale for all.

According to the suggestion, we have converted single channel images to grey-scale.